# Human Pathogenic *Candida* Species Respond Distinctively to Lactic Acid Stress

**DOI:** 10.3390/jof6040348

**Published:** 2020-12-08

**Authors:** Isabella Zangl, Reinhard Beyer, Ildiko-Julia Pap, Joseph Strauss, Christoph Aspöck, Birgit Willinger, Christoph Schüller

**Affiliations:** 1Department of Applied Genetics and Cell Biology (DAGZ), Institute of Microbial Genetics, University of Natural Resources and Life Sciences, Vienna (BOKU), 3430 Tulln an der Donau, Austria; isabella.zangl@boku.ac.at (I.Z.); reinhard.beyer@boku.ac.at (R.B.); joseph.strauss@boku.ac.at (J.S.); 2Institute for Hygiene and Microbiology, University Hospital of St. Pölten, Dunant-Platz 1, 3100 St Pölten, Austria; Ildiko-Julia.Pap@stpoelten.lknoe.at (I.-J.P.); Christoph.Aspoeck@stpoelten.lknoe.at (C.A.); 3Division of Clinical Microbiology, Department of Laboratory Medicine, Medical University of Vienna, 1090 Vienna, Austria; birgit.willinger@meduniwien.ac.at; 4Bioactive Microbial Metabolites (BiMM), Department of Applied Genetics and Cell Biology (DAGZ), Institute of Microbial Genetics, University of Natural Resources and Life Sciences, 3430 Vienna, Austria

**Keywords:** lactic acid tolerance, *Candida*, candidiasis, phenotypic variability

## Abstract

Several *Candida* species are opportunistic human fungal pathogens and thrive in various environmental niches in and on the human body. In this study we focus on the conditions of the vaginal tract, which is acidic, hypoxic, glucose-deprived, and contains lactic acid. We quantitatively analyze the lactic acid tolerance in glucose-rich and glucose-deprived environment of five *Candida* species: *Candida*
*albicans, Candida glabrata, Candida parapsilosis, Candida krusei* and *Candida tropicalis*. To characterize the phenotypic space, we analyzed 40–100 clinical isolates of each species. Each *Candida* species had a very distinct response pattern to lactic acid stress and characteristic phenotypic variability. *C. glabrata* and *C. parapsilosis* were best to withstand high concentrations of lactic acid with glucose as carbon source. A glucose-deprived environment induced lactic acid stress tolerance in all species. With lactate as carbon source the growth rate of *C. krusei* is even higher compared to glucose, whereas the other species grow slower. *C. krusei* may use lactic acid as carbon source in the vaginal tract. Stress resistance variability was highest among *C. parapsilosis* strains. In conclusion, each *Candida* spp. is adapted differently to cope with lactic acid stress and resistant to physiological concentrations.

## 1. Introduction

The human body hosts complex microbial communities [1,2]. Several *Candida* spp. such as *C. albicans* and *C. glabrata* are common human commensals, and are thus highly adapted to humans [3]. They can be found on human skin and mucous membranes like oral or vaginal epithelium and urogenital tract [4,5]. These fungi are opportunistic pathogens, causing mild infections like vulvovaginal candidiasis (VVC) or oral thrush [6], as well as more severe systemic infections in immunocompromised patients. Around 50% of infections are caused by *C. albicans* [7]. Second most common cause is *C. glabrata* with 15–25% of all infections [7,8,9]. Other relevant pathogenic strains are *C. tropicalis*, *C. krusei,* and *C. parapsilosis* [10,11]. Recently, *C. auris* an emerging highly drug resistant species became a focus of concern [12].

*Candida* species are adapted to thrive in the various environmental niches of the human body [13]. Nutrient availabilty, pH fluctuations, and oxygen supply vary greatly between different parts in the human body. For example, pH can be slightly alkaline (pH 7.4) in blood and tissue to acidic (pH 2–pH 6) in gastrointestinal and vaginal tract. In this study we focus on conditions of the vaginal tract, as about 75% of all women suffer of a vaginal *Candida* infection at least once during a liftetime [14]. The environment is glucose deprived, possesses a relatively low pH (pH 4 ± 0.5) accompanied with a up to 110 mM lactic acid and hypoxia [15]. Additionally, it is subjected to regular and developmental environmental changes (e.g., pH rises during menstruation). The vaginal microbiota is dominated by *Lactobacillus* spp., with microbiomes consisting of *L. crispatus*, *L. jensenii*, *L. gasseri,* and/or *L. iners* being the most prominent ones [16,17]. Vaginal lactic acid stems mainly from *Lactobacillus* spp., as human epithelial cells only produce L-lactic acid and the lactic acid in the vaginal tract consists of more than 50% D-lactic acid [18]. However, lactic acid composition and pH slighly depends on the dominating *Lactobacillus* spp. as *L. crispatus* dominated microbiome is generally associated with lower pH and a higher amount of D-lactic acid compared to a *L. iners* microbiome [15,19]. Vaginal *Lactobacillus* species generally differ in their ability to produce lactic acid isomers with *L. iners* only producing L-lactic acid, whereas *L. jensenii* only produces D-lactic acid [19].

Response mechanisms to acid stress differ between *Candida* species. In *C. glabrata* lactic acid response was linked to the hyperosmolarity glycerol response (HOG) pathway and deletion of Hog1 was found to be crucial for tolerance of lactic acid at physiological levels [20]. In glucose-rich environments, *C. albicans* stress response to weak organic acids like lactic acid or acetic acid was reported to be dependent on the transcription factor CaMig1 [21]. Acid stress response mechanisms in other *Candida* species have yet to be investigated.

The role of lactic acid as an antifungal agent is under debate. Lactic acid was reported to effectively reduce *C. albicans* growth [22]. However, a recent report claimed, that physiological concentrations of lactic acid at low pH do not reduce growth of *C. albicans* and *C. glabrata* [23]. Lack of consensus in this point originates possibly also from different growth medium and the used strains. Lactic acid concentration might reach levels higher than 110 mM in vaginal micro milieu or close to a *Lactobacillus* biofilm. *Lactobacillus* spp. are reported to be viable at concentrations of up to 1 M lactic acid at pH 4.5 [24]. The influence of lactic acid against *Candida* species has not been systematically explored yet.

We report here a systematic and quantitative analysis of the effect of lactic acid at low pH on relevant *Candida* species. To avoid phenotypic strain bias, we analyzed for each species small populations of 40 to 100 clinical isolates. We found distinct differences between and within the species populations. Furthermore, since glucose is limited in the vaginal tract, we evaluated also the lactic acid stress response in medium with a nonfermentable carbon source. Additionally, we investigated how efficient different *Candida* species utilize lactic acid at environmental pH as carbon source and found *C. krusei* to deviate from the other species.

## 2. Materials and Methods

### 2.1. Microbial Strains

The clinical *Candida* isolates were collected and provided by the Institute of Hygiene and Microbiology at University Hospital St. Pölten and the Department of Laboratory Medicine at General Hospital Vienna. Isolates were identified by MS, Chromagar *Candida* (BBL™ CHROMagar™ *Candida* Medium, Becton Dickinson GmbH, Heidelberg, Germany) and finally through DNA sequence analysis of the ITS region. A list of the strain’s designations can be found in the Appendix A [25].

### 2.2. Lactate Growth Assay

YNB medium (BD Difco™ Yeast Nitrogen Base without Amino Acids, FisherScientific, Loughborough, UK), supplemented with ammonium sulphate was prepared with 2%(*w/v*) L-lactate (Carl Roth GmbH, Karlsruhe, Germany) or 2% (*w/v*) glucose. pH was adjusted to pH 4 with HCl. *Candida* isolates were grown overnight at 37 °C in YPD (1% yeast extract, 2% peptone, 2% glucose) media in a 96-well flat bottom plate. Using a Hamilton Starlet8 robot (Hamilton Bonaduz AG, Bonaduz, Switzerland) each culture was diluted 1:100 into the respective medium and incubated at 37 °C. Incubation was done in at least triplicates. OD_600nm_ was measured using a fully automated set-up (Cytomat42, Thermo Fisher Scientific, Waltham, MA, USA; Synergy H1 reader, BioTek Instruments Inc., Winooski, VT, USA; Rack Runner 720, Hamilton Bonaduz AG, Bonaduz, Switzerland). Raw data can be found in Appendix A.

### 2.3. Lactic Acid Stress Resistance

Isolates were grown overnight on YPD agar in CELLSTAR^®^ OneWell Plate (Greiner Bio-One GmbH, Kremsmünster, Austria). Each isolate was scraped off and put into 6.5 mL ddH_2_O. Using a Hamilton Starlet8 robot (Hamilton Bonaduz AG, Bonaduz, Switzerland), each isolate was inoculated 1:10 diluted into 200 µL medium per well of standard 96-well flat bottom plates containing a series of L-lactic acid concentrations. Inoculum for each sample was adjusted to OD_600nm_ ~0.04. YP media was prepared with 2% glucose or 2% glycerol (Carl Roth GmbH, Karlsruhe, Germany) as carbon source. Media was supplemented with 80% L−lactic acid solution (Carl Roth GmbH, Karlsruhe, Germany) to achieve the following end concentrations: 160 mM, 320 mM, 480 mM, 640 mM, and 800 mM and pH was adjusted to pH 4. Plates were incubated at 37 °C for 65 h. OD_600nm_ was measured every 2 h using our fully automated set-up. Raw data can be found in Appendix A

### 2.4. Statistical Analysis

All experiments were performed at least in triplicates. All growth rates were calculated using the “growthcurver” package of the statistic software R [26,27]. Graphs were prepared using packages “ggplot2” and “ggstatsplot” [28,29]. Significance testing for *L*-lactic acid stress experiments were performed using the “ggstatsplot” package. Pairwise comparisons were done using a Games–Howell test (Welch’s ANOVA) with Bejamini and Hochberg method for *p*-value adjustment. Significance testing for the cluster analysis was done using Welch’s *t*-test.

## 3. Results

### 3.1. Candida Species Have a Distinct Growth Rate Response to Lactic Acid Stress

In the vaginal tract, *Candida* species are confronted with an overall lactic acid concentration of 110 mM at pH 4 [15]. This concentration could be potentially higher in close proximity to the epithelium cell wall, as it is layered by *Lactobacillus* spp., the main producers of vaginal lactic acid and free diffusion might be reduced by the viscosity of the vaginal fluids and the biofilm. To assess the fitness of different *Candida* species against lactic acid stress, we tested populations of clinical isolates of the main *Candida* species found in the vaginal tract: *C. albicans*, *C. glabrata*, *C. krusei*, *C. parapsilosis*, and *C. tropicalis* [30]. Growth fitness tests were performed by continuous observation of the optical density. Thus, we were able to extract all parameters of the growth curves and use them for analysis of growth parameters. We used several isolates for each species to obtain data on the phenotypic variability. The different responses to lactic acid depending on the *Candida* species are shown in Figure 1. As expected, none of the tested species was inhibited by lactic acid concentrations close to physiological concentration in the vaginal tract, but their growth rate was reduced significantly (Figure 1). The pattern of *C. krusei* isolates represented an exception, as they grew significantly faster at 160 mM lactic acid compared to control with no lactic acid (Figure 1C). Interestingly, growth performance at low lactic acid concentration was generally not indicative for the performance at higher concentrations. For example, *C. krusei* isolates were the most affected strains at 640 mM and 800 mM lactic acid, despite increase in growth rate at 160 mM (Figure 1C). *C. tropicalis* isolates showed high reduction (about 40%) of average growth rate of all species at 160 mM compared to 0 mM lactic acid control but no significant change between 160 mM up to 480 mM lactic acid (Figure 1E). We observed that *C. albicans* isolates had the most uniform reduction of growth rate throughout the different lactic acid concentration tested (Figure 1A).

*C. albicans* was found to be highly susceptible to higher concentrations, experiencing a reduction of growth rate of already 62% at 480 mM (Figure 1A,F). *C. glabrata* isolates showed the highest population variability (Figure 1B), indicating different clusters of strains with specific responses. It was by far the fasted growing *Candida* species, with approximately 50% of all isolates growing faster than all isolates of other species at a concentration of 800 mM (Figure 1B). *C. parapsilosis* was the slowest growing species, but possessed the highest resistance against lactic acid stress with the average growth rate of the population only reduced by 45% at 800 mM lactic acid (Figure 1D). Additionally, the *C. parapsilosis* population appeared to have increase variability at 800 mM lactic acid, whereas other species lose variability at higher concentrations.

### 3.2. Isolates form Clusters with Similar Growth Pattern within Species

Isolates of the same species do not always behave uniformly under environmental influences. To find clusters of different growth behaviour, we analysed the isolates in the upper and lower whiskers at 160 mM, except for *C. parapsilosis* where we grouped the isolates according to growth rate at 800 mM (Figure 2). We expected to see clusters of good performing and bad performing isolates. However, only *C. glabrata* isolates followed this trend (Figure 2B).

Interestingly, *C. krusei* and *C. tropicalis* isolates which had a reduced growth rate at 160 mM, where faster at higher lactic acid concentrations when compared to the faster growing isolates at 160 mM (Figure 2C,E). Some *C. parapsilosis* isolates were fast growing at 800 mM lactic acid but only average at lower lactic acid concentration (Figure 2D). *C. albicans* was the only species without distinct clusters (Figure 2A). We also tested grouping of the isolates according to different lactic acid concentration, with essentially similar results. This is interesting since the genetic difference between *C. parapsilosis* strains is comparably low compared to the *C. albicans* isolates. Therefore, we can conclude that at higher lactic acid concentrations minute epigenetic differences lead to variable phenotypes.

### 3.3. Candida krusei Utilizes Lactate More Efficiently Than Glucose

In the vaginal tract, glucose is not directly available as carbon source for *Candida* species, however, lactic acid is found at a constant concentration. We monitored the growth of *Candida* species on lactate at pH 4, to quantify their ability to utilize lactate at a pH similar to the one in the vaginal tract (Figure 3 and Appendix A).

Overall, *Candida* species grow significantly slower on lactate than on glucose with the stark exception of *C. krusei. C. krusei* isolates had a significantly higher growth rate with lactate compared to glucose. When compared to their respective growth on glucose, *C. parapsilosis* and *C. tropicalis* were found to only have a growth rate reduction at about 50%, making them fairly good lactate utilizer. Despite *C. glabrata* being the fastest growing species on glucose, it had the highest reduction of growth rate on lactate. Whereas growth on glucose shows the isolates had a high variability of maximal growth rate with glucose as corbon source, the differences diminished when grown on lactate. Similar results were obtained with the *C. albicans* isolates tested. Overall, the investigated *Candida* species vary dramatically in their ability to utilize lactate and except for *C. krusei* they are able to utilize glucose more efficiently.

### 3.4. Lactic Acid Response Differs between Glucose-Limited and Glucose-Rich Conditions

Finally, we analysed lactic acid stress in glucose limited conditions to approach the interplay between carbon source and lactic acid stress. We used glycerol as carbon source, as glycerol is naturally present in the vaginal tract [31]. Interestingly, growth rate of all tested *Candida* species was not significantly influenced by 160 mM lactic acid compared to control (Figure 4). *C. krusei* was the fastest growing species (Figure 4F) and again, displayed a higher growth rate at 160 mM, compared to 0 mM (Figure 4C). Interestingly, *C. krusei* average growth rate was reduced by 77% at 320 mM, representing a similar low growth rate compared to other species (Figure 4F). With reduction of growth, the population variability of *C. krusei* was also reduced at 320 mM lactic acid (Figure 4F). *C. albicans* growth rate, which on glucose is effectively inhibited by lactic acid, was similar up to 640 mM (Figure 4A). *C. glabrata* (Figure 4B) and *C. tropicalis* (Figure 4E) had no significant change in growth rate up to 480 mM lactic acid. Surprisingly, no significant growth rate reduction was observed for *C. aparapsilosis* (Figure 4D), making it again the most lactic acid tolerant species. All in all, these results show an increased tolerance to lactic acid when glycerol is present as carbon source.

## 4. Discussion

Lactic acid and low pH are considered to represent important defense mechanisms against bacterial and fungal infections in the vaginal tract [15,32]. However, little is known about the effective range of lactic acid at low pH on *Candida* species usually connected to humans. In this study, we evaluated the tolerance to lactic acid of five relevant *Candida* species (*C. albicans*, *C. glabrata*, *C. krusei*, *C. parapsilosis*, and *C. tropicalis*). In order to analyze the variation between isolates we analyzed populations of isolates collected from the Lower Austria and Vienna region over several years. Our data were generated in a highly standardized manner using automated inoculations and continuous measurements of growth parameters of the treated isolates. Using these methods, we define a characteristic phenotypic space for each *Candida* species interrogated.

Previous studies reported, that physiological concentrations (110 mM) of lactic acid had no effect on *C. albicans* and *C. glabrata* growth [23]. We found a significant decrease in growth rate at the lowest tested concentration (160 mM) but no evidence of a fungicidal effect in our quantitative growth analysis against any isolate (~300 strains) of the tested species at lactic acid concentration up to 480 mM. The epithelium in the vaginal tract is reported to be layered with *Lactobacillus* spp, which is thought to prevent pathogens to adhere to the epithelial cell wall [33]. It is unclear, if lactic acid reaches higher concentrations in close proximity to these *Lactobacillus* spp. biofilms covering the vaginal tract. The increased viscosity due to cervical mucus could decrease diffusion of lactic acid and favor establishment of micro milieus with higher concentrations. Vaginal *Lactobacillus* spp. were reported to grow in lactic acid concentrations of 1000 mM at pH 4.5 [24], showing that high concentrations are not harmful to lactobacilli. Our results show that a mean reduction of growth rate by 50% (MIC50) of *C. albicans* and *C. krusei* isolates required more than 450 mM lactic acid. For *C. krusei*, *C. parapsilosis*, and *C. tropicalis* higher concentrations were necessary to achieve 50% growth reduction. *C. parapsilosis* was the most resistant, as average growth rate was not reduced by 50% with 800 mM lactic acid. A slightly higher MIC50 compared to our results for lactic acid has been reported for strains of *C. albicans*, *C. glabrata* and *C. parapsilosis* [21]. However, the MIC50 values by Cottier et al. [21] are in the upper range of our results, indicating the importance of analyzing isolate populations to detect and characterize the variability of phenotypes and responses.

In support of this, we found a high variability of lactic acid tolerance in different *Candida* populations. Whereas *C. glabrata* formed clusters of fast growing and slow growing isolates, *C. krusei* and *C. tropicalis* isolates growing slower at low lactic acid stress tended to be fast growing at higher concentrations. For *C. glabrata* isolates phenotypic variations were already reported in acetic acid response [34], adhesion, and antifungal susceptibility [35]. These isolates were retrieved from different sites (e.g., bloodstream, vaginal tract, and respiratory tract) collected from patients during diagnostic routine. Therefore, the phenotypic diversity could be explained by selective pressures in the host and differences in the genetic background of the isolates. The genetic plasticity of some Candida species is substantial. For example genetic variations in clonal populations of *C. glabrata* have been suggested to to resulte from selection processes in the human body [36]. The source of the variation could also be an intrinsic property of the species and originating from epigenetic fluctuations.

*Candida* species growing on glycerol as sole carbon source were found to possess an increased stress tolerance to lactic acid (Figure 4). Growth on alternative carbon sources was reported to induce tolerance mechanisms to various stress types in *C. albicans*. [37,38,39]. This is accompanied by physiological changes such as changes in the cell wall composition [40,41,42], which are linked to increased stress tolerance. The vaginal environment consists of approximately 50% D-lactic acid, if dominated by *L. crispatus* [19]. In this study, we used only L-lactic acid. No evidence was reported yet that isomers exhibit a different growth effect on *Candida* species. In *C. alibicans* L- as well as D-lactic acid is taken up by the transporter JEN1 [43]. *C. albicans, C. parapsilosis,* and *C. glabrata* possess orthologs for *DLD1*, a D-lactate hydrogenase in *Saccharomyces cerevisiae* [44]. This hints that they are able to utilize D-lactate, as well as L-lactate. Taken together, this makes it unlikely that different isomers have a different effect.

For *Candida* species the assimilation of different carbon sources is linked to their ability to thrive in several host niches (reviewed in Alves et al. [45]). Human α-amylase present in Lower-Genital-Tract mucosal fluid processes glycogen to support vaginal colonization by Lactobacillus. The maltose and maltotriose produced by α-amylase can be fermented by *C. albicans* but not by the other tested species [46]. In *C. albicans* lactate uptake is facilitated by JEN transporters [43] and used in gluconeogenesis in order to generate hexose and pentose sugars which are needed for nucleotide and cell wall synthesis [47]. In *C. glabrata* L-lactate dehydrogenase Cyb2 is responsible for lactate assimilation in hypoxic conditions like vaginal or gastrointestinal tract [48]. Growth of *C. albicans* and *C. glabrata*, on lactate as carbon source is slower compared to glucose [37,38]. Our results confirmed that also quantitatively. In addition, *C. tropicalis* and *C. parapsilosis* also grew significantly slower on lactate compared to their growth on glucose. In contrast, *C. krusei* reached higher growth rates on lactate than on glucose (Figure 2C) and showed increased growth rates at low amounts of lactic acid on glucose (Figure 1C), as well as on glycerol (Figure 4C). Therefore, *C. krusei* can in vitro utilize lactate as carbon source in presence of glucose or glycerin. Thus, it possesses a unique way to more efficiently utilize lactate compared to other *Candida* species. Our results also show that *C. krusei* has a high tolerance against low amounts of lactic acid. In support of our view, intracellular pH of *C. krusei* only slightly changes when challenged with 106 mM lactic acid undissociated at pH 2.5 [49]. *C.krusei* is a rare cause of refractory vaginitis, but harder to treat than more common *C. albicans* infections as it possesses an intrinsic resistance to fluconazole, a very common antifungal used against vaginal *Candida* infections [14,50].

To fully understand the lactic acid tolerance of *Candida* in the vaginal tract further studies are needed, which take the human microbiome and host interactions into account. However, our results hint that the usual concentration of lactic acid in the vaginal environment is low enough to be used as carbon source by the tested *Candida* species and only has minor part in general defense except for local higher concentrations in the vicinity of lactobacillus biofilms. Thus, lactobacilli perhaps do not generally eliminate *Candida* but merely restrict local growth of them.

In conclusion, we are the first to quantitatively describe L-lactic acid tolerance in different *Candida* species. We show that each *Candida* species not only has a distinct growth response to lactic acid stress, but they also vary greatly in the populations characteristic. Furthermore, we report that lactic acid tolerance is dependent on carbon source in vitro, as *Candida* species are less tolerant at 160 mM lactic acid when grown on glycerol. Lastly, our report is the first to describe *C. krusei* to efficiently use L-lactate as carbon source. We believe that our results contribute to the mechanisms behind the role of lactic acid.

## Figures and Tables

**Figure 1 jof-06-00348-f001:**
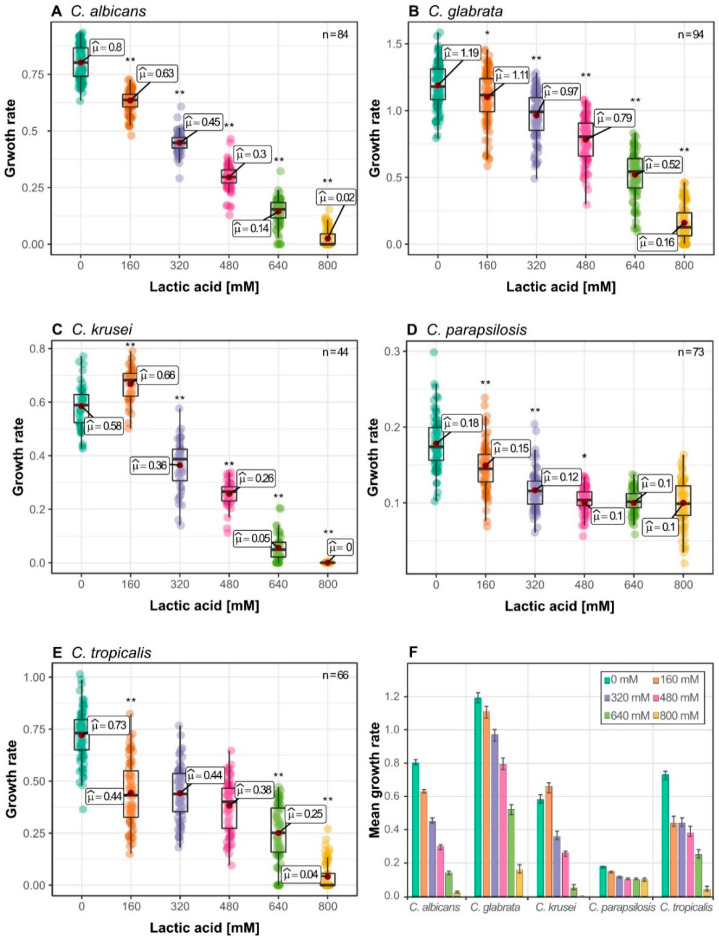
Growth of different *Candida* species under lactic acid stress with 2% Glucose as carbon source: (**A**) *C. albicans*; (**B**) *C. glabrata*; (**C**) *C. krusei*; (**D**) *C. parapsilosis*; (**E**) *C. tropicalis*; (**F**) Mean growth rates of different *Candida* species to lactic acid stress, Error bars represent confidence interval of 95%; Each data point represents mean growth rate of triplicates measurements; Asterisks represents statistical difference of one condition to the next lower lactic acid concentration (* *p* ≤ 0.05, ** *p* ≤ 0.001). Raw data can be found in Appendix A.

**Figure 2 jof-06-00348-f002:**
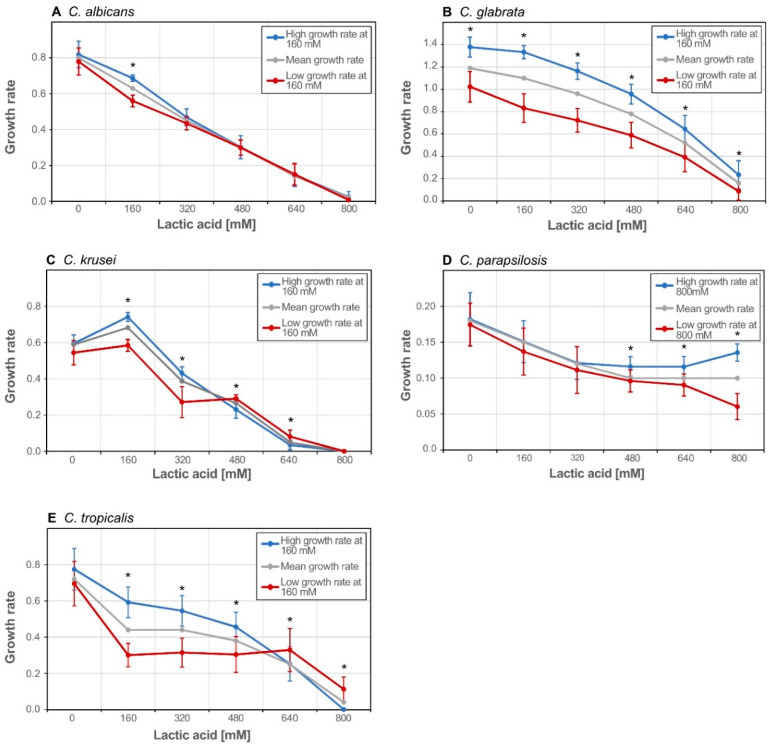
Cluster analysis of lactic acid stress response on Glucose. Isolates were sorted according to their growth rate at 160 mM lactic acid if not stated otherwise and put into high (blue line) and low (red line) growth rate clusters; (**A**) *C. albicans*; (**B**) *C. glabrata*, (**C**) *C. krusei*; (**D**) *C. parapsilosis*, sorted according to the growth rates at 800 mM; (**E**) *C. tropicalis*; Grey line represents the average growth rate per concentration; Error bars represent the standard deviation of the growth rate in a cluster. Asterisks represents statistical difference of one condition to the next lower lactic acid concentration (* *p* ≤ 0.05).

**Figure 3 jof-06-00348-f003:**
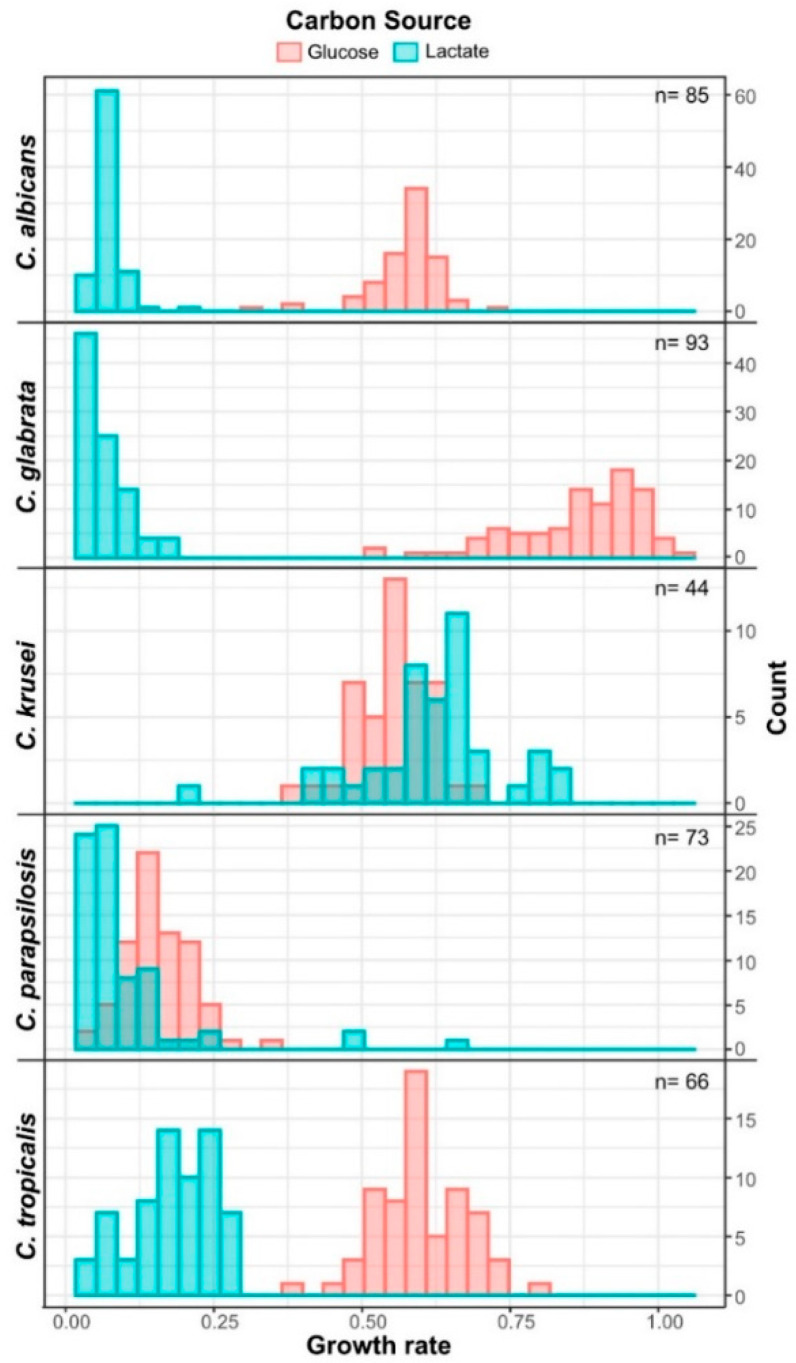
Distribution of growth rate of *Candida* isolates on 2% Lactate or 2% Glucose as sole carbon source at pH 4; Experiment was done in at least triplicates; Mean values of maximal growth rates are reported. Raw data can be found in Appendix A.

**Figure 4 jof-06-00348-f004:**
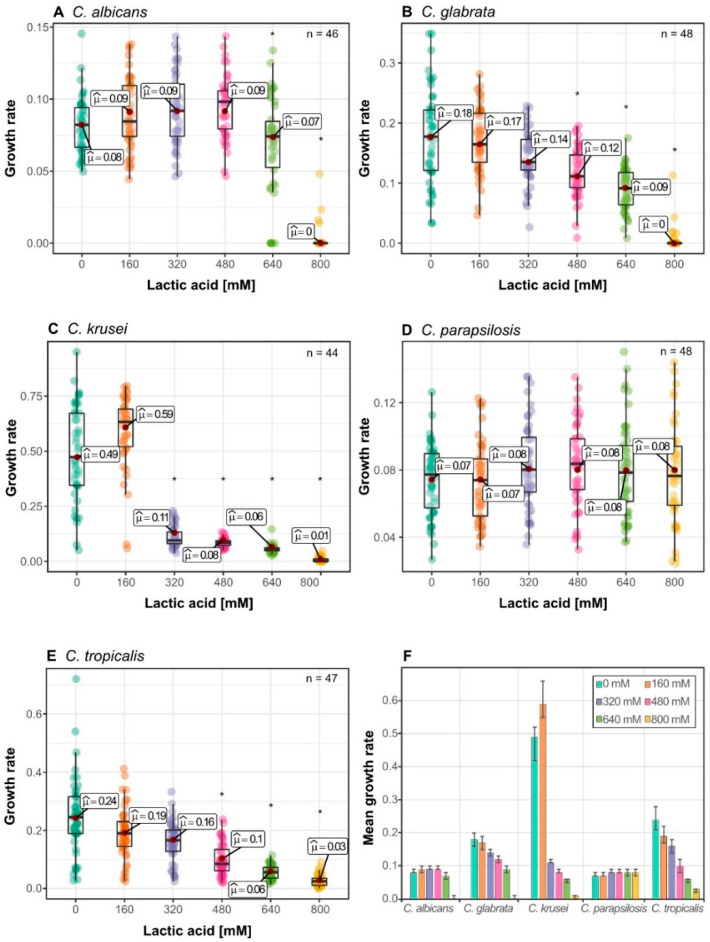
Lactic acid tolerance of *Candida* species using glycerol as carbon source Growth response to lactic acid stress with 2% Glycerol as carbon source; (**A**) *C. albicans*; (**B**) *C. glabrata*; (**C**) *C. krusei*; (**D**) *C. parapsilosis*; (**E**) *C. tropicalis*; (**F**) Mean growth rates of different *Candida* species to lactic acid stress, error bars represent confidence interval of 95%; Each data point represents mean growth rate of triplicates measurements; Asterisks represents statistical difference of one condition to the next lower lactic acid concentration (* *p* ≤ 0.001). Raw data can be found in Appendix A.

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
