# Peer review of "Human Pathogenic Candida Species Respond Distinctively to Lactic Acid Stress"

_jof, 2020, doi:10.3390/jof6040348_

Round 1

Reviewer 1 Report

This is a very clear and concise study of the growth response by various strains of Candida to lactic acid stress that maybe encountered in the environment. Methods are clearly delineated and the results are well presented and easy to follow. Since lactic acid affects cell wall remodeling and subsequent viability and recognition by host defenses, this study is foundational to understanding yeast viability in the host. Introduction and conclusions are well presented and relevant. Overall, a succinct and interesting manuscript. 

Author Response

We thank the referee for this kind statement.

Reviewer 2 Report

The authors of this paper evaluated the tolerance of different Candida species (C. albicans, C. glabrata, C. krusei, C. parapsilosis, and C. tropicalis) to lactic acid. The paper would indeed be of interest for the readership, the methods are adequately described and the results are well presented. The graphs are easy to understand and well visualized.

Taken together I would like to congratulate the authors for this outstanding work. Indeed, I do not have any points to address (which I never did before in a review…).

Apart from some typos that I found (page 2, line 60, "alibicans"; page 3, line 122, Figure 1 no brackets needed; Figure legend 4, D instead of C for C.parapsilosis), and the fact that the recently published review of Sustr et al. (JoF, 2020) might be considered as a reference, I have no other points to address.

Well done!

Author Response

We thank the reviewer for the comments. We checked the mentioned text passages and changed them accordingly. We added the suggested review as reference for vulvovaginal candidiasis (page1, line 37).

Reviewer 3 Report

Invasive candidiasis such as vaginitis can cause significant health issues.  Specific conditions present in the vaginal tract, such as hypoxia and acidity, attracts several groups of bacteria and fungal species. In the study presented large groups of clinical isolates of 5 clinically relevant Candida spp. were tested in the conditions that may represent microenvironment in the host. Data were clustered, analyzed and interpreted and a significant review of the field made.

This study expands the knowledge of specific metabolic features of different species of Candida and presents the genetic diversity associated with each.

 While extensive experimental work was done, there are some assumptions, experimental approaches and interpretations that better could be supported either by the literature or experimentally.  There are ways to improve data processing and presentation.

General and technical comments are presented below.  

General comments:

The authors speculate that C.krusei might use lactic acid as a carbon source in the vaginal tract  (Lines 26, 272-275).  Experiments to prove this point were done under aerobic conditions (section 3.3 and others). At the same time, vaginal microenvironment conditions are described as hypoxic (Line 18).  Is it possible to effectively utilize lactic acid (a non-fermentable carbon source) in hypoxic conditions? The same applies to glycerol (Lines 197-198).

Only L-lactate was used in the experiments, while vaginal D-lactate is >50% (line53).  There is evidence that D- and L- isoforms of lactic acid might be metabolized/tolerated by yeasts in a different way ( for example Pohanka, 2020). While the authors do address these concerns in the discussion (lines 252-257), only transporter JEN1 is mentioned, but no references are given about the metabolic equality of D- and L-forms for Candida spp. A solid reference or simple growth experiment with both isoforms could support the approach taken.

Lactate growth assay:  At the stage when the cultures were suspended from overnight YPD plates into 6.5mL of H2O (lines 96-98), was the cell amount the same across the samples?  Was it assessed and standardized in any way (turbidity, CFU)? If initial CFU load in the microtiter plate was significantly different, this might explain substantial differences in final OD’s interpreted by authors as related to the growth rate. 

 When the effect of lactic acid on growth was tested,  whether average growth increase (C. krusei) or growth reduction was detected (section 3.1), was it species-specific or a result of populational (inter-strain) variability? As an example, the cluster of C. krusei with low growth rates did not show a significant growth increase at 160mM (Fig2C).

The authors conclude that in the case of C. krusei and C. tropicalis  clusters that show slower growth at 160mM  were growing faster at higher concentrations of lactate “… when compared to the average or faster-growing isolates” (lines 154-157, Figures 2C and 2E). Were these differences statistically significant? Based on the size of error bars this is unlikely. 

Growth rate:  In many cases (Figure 3) different isolates of the same strain showed considerable variability in growth rate, even when growing on glucose. As an example, there was an isolate of C. glabrata that had almost no growth on glucose, and some isolates of C. parapsilosis had growth rate on glucose or lactose ~ 7 times higher than the others.  At the same time, data shown at figure 2, where strains were presumably grown at the same conditions (pH4,  2% glucose, at zero lactate point), did not show such variability in the clusters (reflected by error bars).

The statement that  C. parapsilosis species grow significantly slower on lactate than on glucose (Lines 180-182) is controversial.  Similar to   C. krusei,   for C.parapsilosis  it is not clear whether the species, in general, are growing better or worse (Figure 3), as isolate peak distributions do overlap and the individual behaviour of each isolate is not shown. A possible way to illustrate this point is to present the results as a distribution of ratios [glucose growth rate/lactose growth rate] calculated for each isolate.

Lines 285-290:  Exposure to lactate in the environment only can be called as a stress (and trigger stress response) if the concentration exceeds a physiological threshold.  For Candida spp, this might not be the case for most concentrations found in vaginal tract or for those being tested.

Technical corrections:

Line 64: as as

Line 68: higher levels than  -> levels higher than

Lines 229-234: IC50 -> MIC50

Line 270: “ …C. krusei can ferment lactate as a carbon source”. Lactate is non-fermentable carbon source.

Author Response

Point1: The authors speculate that C.krusei might use lactic acid as a carbon source in the vaginal tract  (Lines 26, 272-275).  Experiments to prove this point were done under aerobic conditions (section 3.3 and others). At the same time, vaginal microenvironment conditions are described as hypoxic (Line 18).  Is it possible to effectively utilize lactic acid (a non-fermentable carbon source) in hypoxic conditions? The same applies to glycerol (Lines 197-198).

Response 1: There is not a lot of research done for C. krusei in this field. However, other Candida species like C. glabrata and C. albicans were reported to be able to utilize L-lactate under hypoxic conditions similarly to aerobic conditions (Ueno et al, 2011). We assume that C. krusei would behave similar.

Unfortunately, our experimental set up does not allow anaerobic conditions, thus we are not able to confirm C. krusei using lactate at defined hypoxic conditions.

Point 2: Only L-lactate was used in the experiments, while vaginal D-lactate is >50% (line53).  There is evidence that D- and L- isoforms of lactic acid might be metabolized/tolerated by yeasts in a different way (for example Pohanka, 2020 DOI: 10.1155/2020/3419034). While the authors do address these concerns in the discussion (lines 252-257), only transporter JEN1 is mentioned, but no references are given about the metabolic equality of D- and L-forms for Candida spp. A solid reference or simple growth experiment with both isoforms could support the approach taken.

Response 2: Till today, there are no systematic studies regarding the metabolism of D-lactate in Candida species done but we will take that up in future. However, orthologs of the D-lactate dehydrogenase of S. cerevisiae were found in C. albicans, C. glabrata and C. parapsilosis. To clarify that the text passage was changed accordingly:

Line 255- 257: “Candida albicans, C. parapsilosis and C. glabrata possess orthologs for DLD1, a D-lactate hydrogenase in Saccharomyces cerevisiae [1]. This hints that they are able to utilize D-lactate, as well as L-lactate.”

The reference for S. cerevisiae DLD1 was added:

Lodi, T.; Ferrero, I. Isolation of the DLD gene of Saccharomyces cerevisiae encoding the mitochondrial enzyme D-lactate ferricytochrome c oxidoreductase. MGG Mol. Gen. Genet. 1993, doi:10.1007/BF00291989.

We thank the reviewer for the suggestion. Unfortunately, due to imposed time-constraints we are not in the position to perform the suggested experiment.

Point 3: Lactate growth assay:  At the stage when the cultures were suspended from overnight YPD plates into 6.5mL of H2O (lines 96-98), was the cell amount the same across the samples?  Was it assessed and standardized in any way (turbidity, CFU)? If initial CFU load in the microtiter plate was significantly different, this might explain substantial differences in final OD’s interpreted by authors as related to the growth rate. 

Response 3: We apologize for the confusion. OD600 of the suspension were assessed and, if needed, adjusted to OD600nm~ 0.4 per 6.5ml H2O. After final dilution, starting OD600nm was ~0.04. In addition, experience shows that the maximal growth rate value is robust against variation of the start OD in a relatively wide range.

We added a sentence to clarify that in section 2.3:

Page 3; Line 99:  “Inoculum for each sample was adjusted to OD600nm ~0.04.”

Point 4: When the effect of lactic acid on growth was tested, whether average growth increase (C. krusei) or growth reduction was detected (section 3.1), was it species-specific or a result of populational (inter-strain) variability? As an example, the cluster of C. krusei with low growth rates did not show a significant growth increase at 160mM (Fig2C).

Response 4: Our assays were done in triplicates and the variation between the samples was overall less than 5% and these data was and is included in the supplementary file S1. Thus we concluded a species specific profile from our measurements.

Point 5: The authors conclude that in the case of C. krusei and C. tropicalis  clusters that show slower growth at 160mM  were growing faster at higher concentrations of lactate “… when compared to the average or faster-growing isolates” (lines 154-157, Figures 2C and 2E). Were these differences statistically significant? Based on the size of error bars this is unlikely.

Response 5: Statistical analysis (Welch’s t-test) between “high” growing and “low” growing strains show a significance between both populations from 160mM to 800mM for C. tropicalis and from 160mM to  640mM (p < 0.05).

The sentence was changed to “Interestingly, C. krusei and C. tropicalis isolates which had a reduced growth rate at 160 mM, where faster at higher lactic acid concentrations when compared to the faster growing isolates at 160mM (Figure 2C and 2E).” (page 5, line 155-157)

P values were added to Figure 2.

One sentence in Section 2.4 was added to explain significance testing method for cluster analysis. Line 109-110:“Significance testing for the cluster analysis was done using Welch’s t-test.”

Point 6: Growth rate: In many cases (Figure 3) different isolates of the same strain showed considerable variability in growth rate, even when growing on glucose. As an example, there was an isolate of C. glabrata that had almost no growth on glucose, and some isolates of C. parapsilosis had growth rate on glucose or lactose ~ 7 times higher than the others.  At the same time, data shown at figure 2, where strains were presumably grown at the same conditions (pH4,  2% glucose, at zero lactate point), did not show such variability in the clusters (reflected by error bars).

Response 6: We apologize for the inconsistency. The data for Figure 3 was measured with different media (as describe in Materials and Methods) than the lactic acid stress resistance. For lactate as carbon source, YNB, a defined synthetic medium had to be used, whereas for the stress testing the YPD medium was used. Generally, we observed that growth rate on YNB is less compared to growth rate on YPD.

We checked data of all isolates initially and reevaluated extremes outliers yet these escaped our filters. We thank the referee for his most careful look at our results.

The following strains were re-examined: 346B (C. parapsilosis according to our ITS results- removed), 1379W (C. glabrata but strangely slow growing - > removed), 82I (C. krusei but mislabeled as C. tropicalis -> was merged with the C. krusei data)

All Figures and Data were adjusted accordingly.

Point 7: The statement that C. parapsilosis species grow significantly slower on lactate than on glucose (Lines 180-182) is controversial.  Similar to   C. krusei,   for C.parapsilosis  it is not clear whether the species, in general, are growing better or worse (Figure 3), as isolate peak distributions do overlap and the individual behaviour of each isolate is not shown. A possible way to illustrate this point is to present the results as a distribution of ratios [glucose growth rate/lactose growth rate] calculated for each isolate.

Response 7: The data has been submitted as part of the supplementary data in the original submission. Nevertheless, we do now show the requested data as distribution in a new Supplementary Figure 1S.

Point 8: Lines 285-290:  Exposure to lactate in the environment only can be called as a stress (and trigger stress response) if the concentration exceeds a physiological threshold.  For Candida spp, this might not be the case for most concentrations found in vaginal tract or for those being tested.

Response 8: We agree with the referee. The text passages were changed accordingly:

line 287: the word “stress” was deleted

Technical corrections:

Line 64: as as -> changed

Line 68: higher levels than  -> changed to “ levels higher than”

Lines 229-234: IC50 -> changed to MIC50

Line 270: “ …C. krusei can ferment lactate as a carbon source”. Lactate is non-fermentable carbon source.

The sentence was changed to “C. krusei can grow with lactate as sole carbon source”